# Temperature-dependence of early development of zebrafish and the consequences for laboratory use and animal welfare

Angelina Miller[1]☉*, Katja Lisa Schröder[1]☉*, Karsten Eike Braun[1], Caitlin Steindorf[1], Richard Ottermanns[1], Martina Roß-Nickoll[1], Henner Hollert[2,3], Thomas Backhaus[1,4]

**1** Institute for Environmental Research, RWTH Aachen University, Aachen, Germany, **2** Department for Evolutionary Ecology and Environmental Toxicology, Goethe University, Frankfurt am Main, Germany, **3** Department of Environmental Media-related Ecotoxicology, Fraunhofer Institute for Molecular Biology and Applied Ecology, Schmallenberg, Germany, **4** Department of Biological and Environmental Sciences, University of Gothenburg, Gothenburg, Sweden

☉ These authors contributed equally to this work.
* angelina.miller@ifer.rwth-aachen.de (AM); katja.schroeder@ifer.rwth-aachen.de (KLS)

## Abstract

Zebrafish (*Danio rerio*) are widely used in biological research, but the impact of incubation temperatures on developmental endpoints is still insufficiently studied. This study quantifies developmental differences in zebrafish embryos incubated at 26°C and 28°C, focusing on key endpoints (heartbeat onset, hatching time, eye size, yolk sac consumption, and body length). For this purpose, we recorded a high-resolution time series comprising hourly observations of early developmental stages and key events and bi-hourly observations of body length until 120 hours post fertilization. Additionally, we recorded a low-resolution time series at 72, 96, and 119 hours post fertilization for detailed measurements of eye size, yolk sac area, and body length. Embryos incubated at 26°C showed consistent delays in developmental stages compared to those at 28°C, with delays becoming more pronounced at later stages. Yolk sac consumption was delayed by about 19.8 hours at 26°C by 119 hours post fertilization, suggesting a delayed onset of independent feeding. These findings suggest that time-based regulatory limits for rearing zebrafish, such as the 120-hour threshold in German regulations (TierSchVerV), do not fully account for temperature-dependent development. The results emphasize the need for guidelines linking incubation temperatures to developmental progress.

## 1. Introduction

The zebrafish, *Danio rerio*, is a widely used animal model in biological and biomedical research [1,2], valued for the homology of many of its organs and their development with other vertebrates, including humans [3,4]. Therefore, experimental conditions are largely standardized either through legal requirements such as the EU Directive

**Data availability statement:** All datasets as well as the R (version 4.3.3) code used are provided as comma separated plain text value files and are publicly available on Zenodo (https://doi.org/10.5281/zenodo.13905257).

**Funding:** This work was funded by the Deutsche Forschungsgemeinschaft (DFG, German Research Foundation) under Germany´s Excellence Strategy – Cluster of Excellence 2186 „The Fuel Science Center" – ID: 390919832.
Funded by the Verein des Hygiene-Institut des Ruhrgebiets e.V., Gelsenkirchen, Germany, funding contract "Standardised and automated evaluation of the zebrafish behaviour test for the environmental toxicological assessment of pollutants" (phase 1 and 2). The funders had no role in study design, data collection and analysis, decision to publish, or preparation of the manuscript.

**Competing interests:** The authors have declared that no competing interests exist.

2010/63 [5], individual countries, e.g., in the German regulation TierSchVerV [6] or on the basis of widely cited analyses published in the scientific literature [7–10]. These guidelines define good practice for zebrafish maintenance and laboratory use and form the basis for the standardized assessment of various zebrafish characteristics.

However, the optimal thermal conditions for the maintenance and laboratory use of zebrafish are still subject of considerable debate. While OECD and DIN EN ISO standards recommend temperatures of $26 \pm 1°C$ [11,12], the most frequently used maintenance temperature is 28.5°C [13]. Other scientific publications favor incubation temperatures ranging from 26°C to 28°C [14].

Furthermore, the developing fish embryo is the most sensitive stage in the fish life cycle [15]. This stage is, therefore, particularly vulnerable to thermal stress, as embryos possess limited capacity to regulate membrane fluidity, adjust metabolic rates, or engage in behavioral thermoregulation [16–18].

The influence of temperature is particularly critical when analyzing endpoints that can only be recorded shortly before the zebrafish larvae enter their protected status according to EU Directive 2010/63 [5], such as locomotor behavior and the conclusions of the validation efforts for the fish embryo toxicity (FET) test under the OECD [10]. According to 2010/63 and as detailed in Strähle et al. [14], early, non-"free-living" developmental stages of laboratory animals are not classified as vertebrates and are therefore not subject to special protection. Such developmental stages of fish are defined in the Directive as all stages that do not feed independently, even if already hatched, independent of the incubation temperature. Although Braunbeck et al. [10] estimate that zebrafish eleutheroembryos do not feed independently prior to 120 hours post fertilization (hpf), they do not define the underlying incubation temperature. In a follow-up paper by Strähle et al. [14], the authors recommend 120 hpf as a safe limit for incubation at 28.5°C and use an equation earlier developed by Kimmel et al. [8] to extrapolate to an incubation time of 139.5 hpf at 26°C before independent feeding starts. This equation was developed from staging data up to 72 hpf of three incubation temperatures (25°C, 28.5°C, 33°C) and assumes a linear relationship between fish development rate and temperature.

However, growth rates in biological systems typically follow a non-linear relationship that peaks at an optimum temperature [19], which was also demonstrated for fish growth processes [20]. We therefore hypothesize that the current estimation of the maximum incubation duration before zebrafish eleutheroembryos commence independent feeding (and thereby enter into a life stage in which they enjoy special protection as vertebrates) is not sufficiently precise for extrapolation to other temperature conditions than 28°C.

The aim of this study was to quantify the differences in zebrafish development that result from two incubation temperatures (26°C and 28°C) and to put them into the context of the considerations put forward by EU Directive 2010/63, Braunbeck et al [10] and Strähle et al [14]. For this purpose, we analyzed the development characteristics in early embryonic stages, heartbeat onset and hatching time, as well as body length in detail using a twice repeated high-resolution time series, which was amended by a low-resolution time series (repeated thrice), in order to record details

on eye and yolk sac size. This data can help responsible stakeholders in animal welfare make an informed decision on time limits for the use of zebrafish embryos dependent on experiment temperature.

## 2. Methodology

Our experiments were not considered animal testing according to EU Directive 2010/63 [5] as all animals were euthanized before 120 hpf.

### 2.1 High-resolution time series

All details of the set-up of the time series are presented in Miller et al. [21]. Briefly, we transferred 24 embryos of wild-type zebrafish (*Danio rerio)* per temperature group (26°C and 28°C) to 96-round-well-flat-bottom plates with 300 µL of embryo medium and photographed them individually every hour using a digital microscope (Keyence VHX-970F, Neu-Isenburg, Germany). After all 24 embryos were hatched, we photographed them every two hours until 119 hpf.

We used data from two independent experiments (based on different egg batches obtained from a separate spawning event), resulting in data from 47–48 individuals per temperature group, where the individuals served as independent replicates.

**2.1.1 Classification of early developmental stages.** Based on the classification by Kimmel et al. [8] we identified developmental stages between 1 hpf and approximately 40 hpf from the high-resolution time series (supplementary information, S1 Table). For accurate classification, we used the primary and easily recognizable developmental characteristics of each developmental stage. If an individual could not be clearly assigned to the next higher developmental stage, we classified it under the previous stage.

### 2.2 Low-resolution time series

The detailed methodology for the low-resolution time series is presented in Miller et al. [21]. Briefly, after 72, 96 and 119 hours of incubation and anesthesia in 0.4 g benzocaine/L $H_2O$ ($C_9H_{11}NO_2$, CAS #94-09-7) for at least ten minutes, we individually adjusted about 24 eleutheroembryos of wild-type zebrafish (*Danio rerio*) per temperature group (26°C and 28°C) laterally in a gelatinous fixative (1.67 g dry gelatine/50 mL $H_2O$). This optimized the overlap of the two eyes, thereby facilitating the recording of an image that enables the quantitative comparison of the eleutheroembryo's dimensions. In this experiment, we quantified the area of the top-facing eye, and the yolk sac size using the microscope software's internal scale.

We repeated the experiment twice, resulting in data from 53–61 successfully fixated individuals, per temperature and time group (72 hpf, 96 hpf, 119 hpf). The individuals are considered as independent replicates and the three experiments originated from different egg batches obtained from spawning events at least one week apart.

### 2.3 Statistical analysis

We performed all statistical analyses with R [22] in the RStudio IDE [23] and used base R or the package *ggplot2* [24] for plotting the data. Data was cleaned to reduce the impact of measurement uncertainties by applying the 1.5 inter quantile range rule per time point (unless n < 4). We chose to rely on non-parametric bootstrapping methods for delay estimation in order to obtain robust confidence intervals.

We analyzed the experiments separately, in order to elucidate variability between the experimental runs for which the egg batches were collected at different days. Detailed results for all experiments, together with the corresponding coefficients of variation, are provided in S2 Table of the supplementary information.

**2.3.1 Early developmental stages.** Because the staging data are not normally distributed, we assessed differences in development speed with a permutation test (2000 resamples) implemented via the *independence_test* function from the coin package in R [25]. Replicate affiliation was entered as a blocking factor to control for batch effects, and p-values were

adjusted for multiple comparisons with the Bonferroni correction. The magnitude of the delay was summarized as the median difference, with 95% bootstrap confidence intervals calculated from resamples that were confined within replicates (2000 resamples).

**Comparison with Kimmel et al.:** We compared our data with the estimates provided by the equation of Kimmel et al. [8] which relates development velocity to development at 28.5°C as follows:

$$H_t = \frac{h}{0.055 * T - 0.57}$$

(1)

With $H_t$ = hours of development at temperature $T$, $h$ = hours of development at 28.5°C and $T$ = temperature in °C. We also re-evaluated the equation, using Kimmel et al.'s own data series for 25°C, 28.5°C and 33°C, retrieved from the paper with the help of the web plot digitizer [26].

**2.3.2 Onset of heartbeat and hatching.** We tested the time to hatching and the onset of heartbeat for statistically significant difference, analogous to the early developmental stages (2.3.1). To visualize the relationship between incubation duration and the fraction of embryos having a heartbeat we used a log logistic function (2 parameters) with package *drc* [27].

**2.3.3 Body length.** Growth patterns generally follow a logistic growth curve [28]. However, our data represent only a particular segment of this curve. Under this condition, the precise shape of the growth curve that is to be modelled can have a linear (middle part of a logistic curve) or a non-linear shape (early and later part of a logistic curve). We therefore applied a Shape Constrained Additive Model (SCAM) with integrated smoothness [29]. This approach ensures the fitted model adheres to biologically plausible constraints, specifically, a monotonically increasing trend with a concave curvature if supported by the data, and can take a linear or non-linear shape, depending on the experimental data. To account for variability between experiments (batch effects), we incorporated a random effects term into the model.

To quantify the delay in hpf between the temperatures at the last measured time point (118 hpf), we determined the time when eleutheroembryos incubated at 26°C reached the same length as the ones incubated at 28°C. To estimate the confidence interval for the developmental delay, we used a bootstrap approach in which the models were refitted 5000 times to bootstrap samples (resampled with replacement from the original data). The 95% interval of sampling distribution was used as the confidence interval. To evaluate the statistical significance of the observed delay patterns, we conducted 2000 permutation resamples in which the temperature labels were randomly shuffled. For each resample, the model was refitted, and the resulting delay estimate was recorded. The p-value was then calculated as the proportion of permutation delays that exceeded the empirical delay.

To estimate the impact of repeated handling of the embryos on growth, we compared the final lengths of the hourly to bi-hourly measured eleutheroembryos from the high-resolution time series with the ones measured in undisturbed individuals from the low-resolution time series.

To assess the optimal temperature range for embryonic development, body length data from the low-resolution time series were used as a proxy for embryonic development and plotted against the corresponding incubation temperatures. In addition to the two temperatures presumed to lie within the optimal range (26°C and 28°C), we also investigated body length development at more extreme temperatures. For this purpose, an additional low-resolution time series was conducted at 24°C and 30°C, and body lengths were measured accordingly. The detailed methodology is described in Miller et al. [21]. Briefly, the experiment was repeated three times under each temperature condition (24°C and 30°C), resulting in data from about 72 independent individuals per temperature. The three egg batches served as independent experiments. We tested for nonlinearity in the relationship between body length and incubation temperature using a quadratic regression model, with temperature included as both a linear and a squared term. The replicate label was initially included as a fixed factor to account for possible differences among egg batches, but as replicate effects were not significant, the

term was removed for the final model comparison. The quadratic model was also compared against a linear model, using an F-test and Akaike's Information Criterion (AIC) for significance testing.

**2.3.4 Yolk sac and eye size.** Data for the areas of the yolk sac and eye size are available at three time points: 72, 96, and 119 hpf, with n = 176 at 26°C for both endpoints and n = 174 and 175 at 28°C, respectively. To compare the delay that occurred at 119 hpf, we applied the same modeling strategy as described previously for body length data (2.3.3) to the eye size data. For yolk reduction, we applied a monotonously decreasing spline with convex shape, because the time frame of our measurements lies in the linear to lower half of the inverted S-shape of the logistic decay that has been described earlier [28].

## 3. Results and discussion

To assess and quantify differences in development speed of zebrafish embryos, incubated at 26°C and 28°C, we analyzed the early developmental stages, onset of heartbeat, hatching as well as body length, eye size and yolk sac size over time.

### 3.1 High-resolution time series

**3.1.1 Early developmental stages.** Fig 1 shows the time (hpf) zebrafish embryos need to reach a specific developmental stage at an incubation temperature of 26°C or 28°C. The early cell stages (4-cell (4c) to sphere (S)) of the first few hours in development show no difference between the two incubation temperatures. From the shield (SH) stage on all stages are consistently reached significantly faster at 28°C than at 26°C with a CI of median difference not adjacent to zero (all p < 0.0001). Differences in developmental progress at 14, 18, and 24 hpf are presented in S3 Table of the supplementary information. At these early stages, somite number serves as a reliable visual marker for assessing

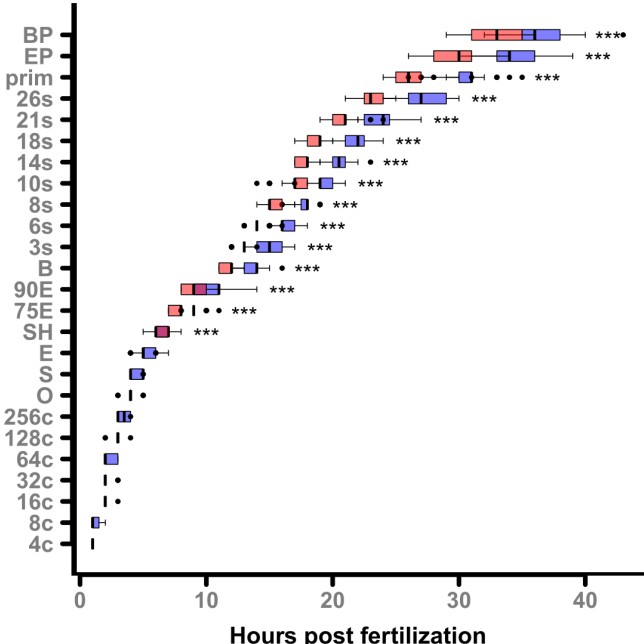

**Fig 1. Time to reach developmental stages at 26 and 28°C.** Hours post fertilization (hpf) for various developmental stages at 26°C (blue) and 28°C (red). Data is shown as upper and lower quartiles (boxes), median (middle line), minimum and maximum values (bars), and outliers for each stage (dots) at both temperatures; n_26°C = 47, n_28°C = 48. Statistical significance according to permutation test (α = 0.05) is given as * p < 0.05, ** p < 0.01 and *** p < 0.001. The explanations of the abbreviations are given in S1 Table of the supplementary information.

developmental delay, while pigmentation becomes a more prominent indicator at slightly later stages. Beyond these time points, distinct visual cues become less pronounced, necessitating the use of more detailed morphological endpoints to accurately quantify developmental differences.

At around 1.5 days post fertilization (dpf) the embryos incubated at 26°C have a median developmental delay of at least 3.25 hours (95% CI 3.25–5). S4 Table in the supplementary information presents a detailed statistical analysis of the developmental stages, including the sample sizes (n), the differences in means, the bootstrapped CI of difference in medians and the p-values indicating the significance of the differences observed.

S2 Table in the supplementary information shows the differences at the end of staging (body pigmentation (BP)) for the temperatures separated by replicate. The CIs for differences in medians overlap (replicate 1: 95% CI 3–4.5; replicate 2: 95% CI 1–5).

**Comparison of findings with Kimmel et al.:** When comparing our development data with the data and equation derived by Kimmel et al. [8] it becomes clear that our data have an approximately 15% lower developmental rate (supplementary information, S4 Fig). We attribute this to the different experimental set-ups: while we raised the embryos in 96-round-well-flat-bottom plates and removed them from the incubators for about 15% of total time for the measurements at room temperature, Kimmel et al. [8] incubated the embryos in 150 mL beakers from which they were only removed for approximately 8% of the total incubation time, resulting in a higher developmental rate compared to our results. Also, the exposure system itself may contribute to the observed differences in developmental rates, as factors such as water volume, surface-to-volume ratio, gas exchange, and thermal buffering differ between well plates and beakers.

All these findings highlight the importance of minimizing handling time and carefully considering and documenting the physical characteristics of the exposure system. To ensure accurate interpretation of experimental outcomes it is essential to apply consistent handling protocols across all groups and to include appropriate controls to account for handling-related effects when working with well plates. Additionally, genetic differences between the different laboratory strains could contribute to differences between laboratories and temporally distant experiments, as gene drift can occur over time [30].

**Reanalysis of Kimmel et al. data:** The data we derived from the original plot indicate that the slopes predicted by Kimmel's formula deviate from the actually observed slopes by at least 2% at 25°C and at least 10% at 33°C (supplementary information, S5 Fig and S5 Table). Based on our re-analysis of the data on zebrafish development that were recorded by Kimmel et al. [8] at 25°C, the predicted development rate is 0.80, estimated using Eq. 1. The slope of the fitted regression line for this temperature is 0.77 ± 0.02. The experimental data (supplementary information, S5 Fig, light blue) and the predicted values (supplementary information, S5 Fig, dark blue line) show that the actual development rate is slightly slower than predicted. At 33°C, the predicted development rate is 1.24. The slope of the fitted line here is 1.11 ± 0.02. The observed data points (supplementary information, S5 Fig, orange) and the predicted values (supplementary information, S5 Fig, red line) indicate that the development rate at 33°C is also slightly lower than the model's prediction.

Overall, the data show that while the development rate increases with temperature, as reflected in the steeper slopes of the lines at higher temperatures, the observed rates are consistently lower than predicted. This suggests that the model used for prediction overestimates development rates, particularly at the more extreme temperatures of 25°C and 33°C. The variation in the slopes and their uncertainties indicates that the relationship between temperature and development rate might require further refinement for more accurate predictions.

These deviations support our hypothesis that the relationship between temperature and growth rate is not linear but follows an optimum curve. The more significant downward deviation at 33°C suggests that this temperature is further from the optimal range, which is likely around 28.5°C. We found increased mortality rates of 19% at 30°C and 35% at 24°C incubation temperature after 120 hpf. Additionally, we observed pericardial edema at a rate of 39% for 24°C and 14% at 30°C. This is consistent with findings from Pype et al. [31], who reported an increased rate of malformations at temperatures above 32.5°C, further indicating that 33°C is outside the ideal range for normal development.

 

**3.1.2 Onset of heartbeat.** The median onset time of the heartbeat at 28°C is 25 hpf, which is 4 hours (95% CI 3.5–5) earlier than the median onset time at 26°C (Fig 2) (permutation test p < 0.0001). S2 Table in the supplementary information shows the differences for the temperatures separated by replicate. The two replicates show a clear overlap of their confidence intervals (replicate 1: 95% CI 3–5, replicate 2: 95% CI 4–6).

**3.1.3 Hatching.** The median hatching time was 68 hpf at 26°C which is significantly longer (p < 0.0001, permutation test) than the 60 hpf hatching time at 28°C by 8 hours (95% CI 2.25–9) (Fig 3). S2 Table in the supplementary information shows the differences for the temperatures separated by replicate. The two replicates showed an overlap in CIs (replicate 1: 95% CI 3.5–11, replicate 2: 95% CI 1–14).

**3.1.4 Body length.** In general, the body length of eleutheroembryos from the high-resolution time series was smaller than those from the low-resolution time series after 119 hpf (by 0.48 mm when comparing the 26°C data and by 0.43 mm when comparing the 28°C data). This is likely caused by the more constant temperature and handling profile of the eleutheroembryos from the low-resolution time series as those were only taken out of the incubators twice during the experiment, corresponding to a maximum of approximately 2% of the total duration of 120 hpf. In contrast, embryos in the high-resolution time series were exposed to more frequent handling, being removed from the incubator for hourly to bi-hourly measurements at room temperature, resulting in an estimated 15% of the total time spent outside the incubator. These findings suggest that not only the type of exposure system (e.g., well plate vs. beaker, see also 3.1.1), but also the cumulative duration that embryos spent outside the incubator, significantly influence developmental progression. Additionally, the time outside the incubator was used for imaging, which may have induced phototoxic effects, as demonstrated by Villamizar et al. [32], who reported that larvae exposed to constant white light exhibited the lowest growth rate at 5 days posthatching. Continuous light exposure is known to disrupt normal circadian clock development and interfere with clock-controlled physiological processes [33]. Therefore, increased light exposure during imaging,

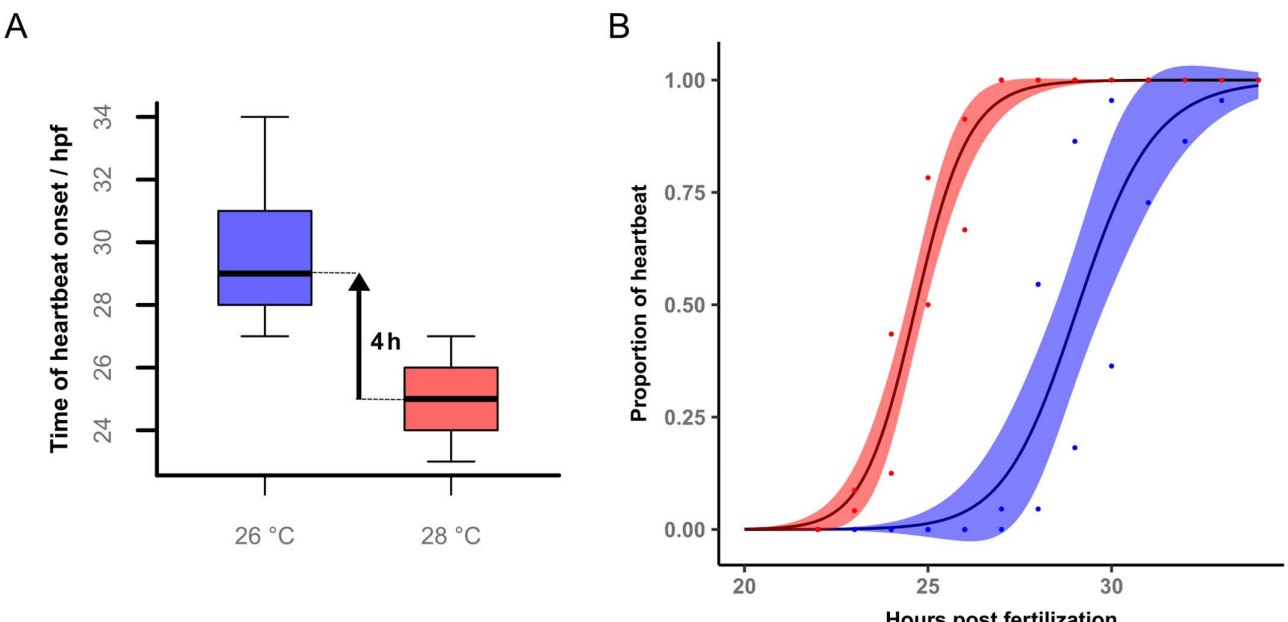

**Fig 2. Heartbeat onset 26 and 28°C. (A)** Time of heartbeat onset (hpf) at 26°C and 28°C shown as boxplots. **(B)** The proportion of embryos exhibiting heartbeat over development time modelled as log-logistic function (hpf) at 26°C (blue) and 28°C (red). The shaded areas represent the 95% confidence intervals ($n_{26°C} = 47$, $n_{28°C} = 48$ in two repeats; p < 0.0001, permutation test).

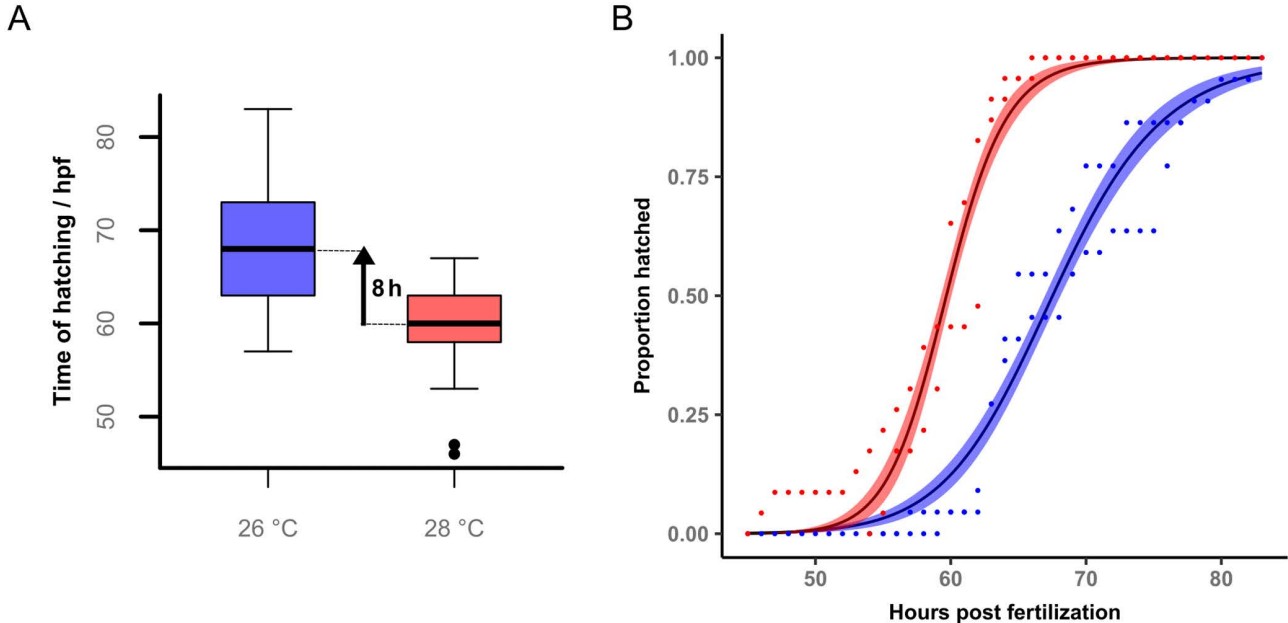

**Fig 3. Hatching onset at 26 and 28°C. (A)** Time of hatching (hpf) at 26°C and 28°C shown as boxplots. **(B)** The proportion of embryos hatched over development time (hpf) at 26°C (blue) and 28°C (red) modelled with a log-logistic function. The shaded areas represent the 95% confidence intervals ($n_{26°C} = 45$, $n_{28°C} = 46$ in two repeats; $p < 0.0001$, permutation test).

particularly during periods corresponding to the dark phase, could also have contributed to the reduced body lengths observed in our study.

Batch variation also differs between designs: in the low-resolution time series, replicate delays are very similar (maximum deviation 2.2 hours; see supplementary information S1 Fig), whereas in the high-resolution time series, replicates differ by 18.6 hours. However, the overall average for the high-resolution time series still aligns reasonably well with the low-resolution time series result. This suggests that frequent handling over five days may have increased batch differences.

Minimizing this time is therefore critical for maintaining stable conditions in temperature-sensitive developmental studies. Based on these findings, we recommend conducting preliminary experiments to assess potential handling, phototoxicity, and batch effects.

Fig 4 illustrates body length growth over time at both incubation temperatures and for both experiment variants high-resolution (imaged every 2 hours in well plates) and low-resolution (imaged while fixated laterally at 72, 96 and 119 hpf).

In the high-resolution time series eleutheroembryos incubated at 28°C reached a median body length at 119 hpf of 3.46 mm. In contrast, eleutheroembryos incubated at 26°C had a median body length of 3.29 mm. The average delay in body growth of the eleutheroembryos incubated at 26°C is 18.6 hpf (95% CI 15.9–21.3, $p < 0.001$) at 118 hpf. For the low-resolution time series a mean delay of 12.9 hpf (95% CI 7.9–17.6, $p < 0.001$) was found. Due to fewer measured time points, the CI is wider than in the high-resolution time series. While the variation in individual body length is consistently larger at 28°C incubation, the estimated average delays for the two experiments are close, differing by only 5.7 hours. In the high-resolution time series, however, the lower bound of the confidence interval suggests a slightly larger delay (7.7 hours difference). Another indication of advanced development is that, while the 26°C curve remains approximately linear, the 28°C curve approaches the saturation phase of logistic growth towards the end of the measurement period (approximately 110 hpf).

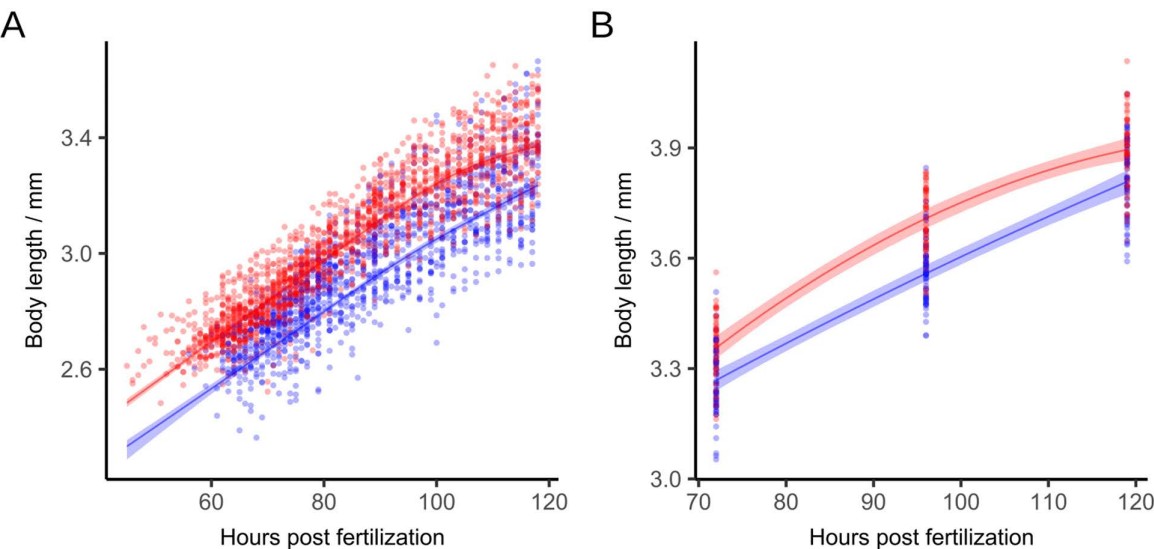

**Fig 4. Body length over time.** Body growth of zebrafish eleutheroembryos over development time in hpf at 26°C (blue) and 28°C (red). A Shape Constrained Additive Model (monotonously increasing and concave bend) was fitted with a random effect term for replicate affiliation. The shaded areas represent the 95% confidence intervals. **(A)** For the high-resolution time series, the points represent measurements in a 96-well plate where individuals were in a good enough position for their length to be measured ($R^2_{26°C}$ = 0.79, $R^2_{28°C}$ = 0.86; $n_{26°C}$ = 45, $n_{28°C}$ = 46 in two repeats; p < 0.001, permutation test). **(B)** For the low-resolution time series different individuals were fixated laterally to be measured ($R^2_{26°C}$ = 0.85, $R^2_{28°C}$ = 0.86; $n_{26°C}$ = 172, $n_{28°C}$ = 170 in three repeats; p < 0.001, permutation test).

**Optimum curve for body length:** Fig 5 presents the relationship between development rate (measured as body length at approximately 120 hpf) and temperature based on the low-resolution time series. At 120 hpf, the mean body lengths were 3.47 ± 0.13 mm at 24°C, 3.80 ± 0.12 mm at 26°C, 3.90 ± 0.12 mm at 28°C, and 3.79 ± 0.14 mm at 30°C. The quadratic model provides a significantly better fit than the linear model (F = 178.17, p < 0.0001; ΔAIC = 135), indicating a curvature in the temperature–length relationship within the observed time interval. Although batch effects were expected biologically, they were not statistically significant, possibly due to small effect sizes relative to residual variation. Residuals were approximately normally distributed (supplementary information, S2 Fig). Length increases from lower to higher temperatures, reaches its maximum at 28°C, and declines again for 30°C. While this pattern is consistent with an optimum-shaped response, further measurements beyond the current temperature range would be required to fully characterize the shape of the thermal performance curve.

On the basis of using staging and body length data as proxies, we conclude that the temperature dependency of zebrafish development does not follow a linear trend in this temperature spectrum, as previously suggested by Kimmel et al. [8] and adopted for extrapolation by Strähle et al. [14].

### 3.2 Low-resolution time series

**3.2.1 Yolk sac consumption.** The measurements of 72, 96 and 119 hour old eleutheroembryos show a continuous consumption of the yolk sac with progressive development. See S3 Fig for regression coefficients and residual plots. Yolk sac degradation is faster at 28°C (Fig 6, the curves are moving apart). As a result, the eleutheroembryos incubated at 26°C are delayed in their development by 19.8 hpf at the end of the incubation (95% CI 13.4–26.3, p < 0.001).

Both temperature groups of eleutheroembryos had not fully consumed their yolk sacs after 119 hours, indicating ongoing reliance on yolk as an endogenous energy source. This aligns with previous findings by Kalasekar et al. [34], who found that the eleutheroembryos' functional gut develops at approximately 120 hpf, initiating external food ingestion while

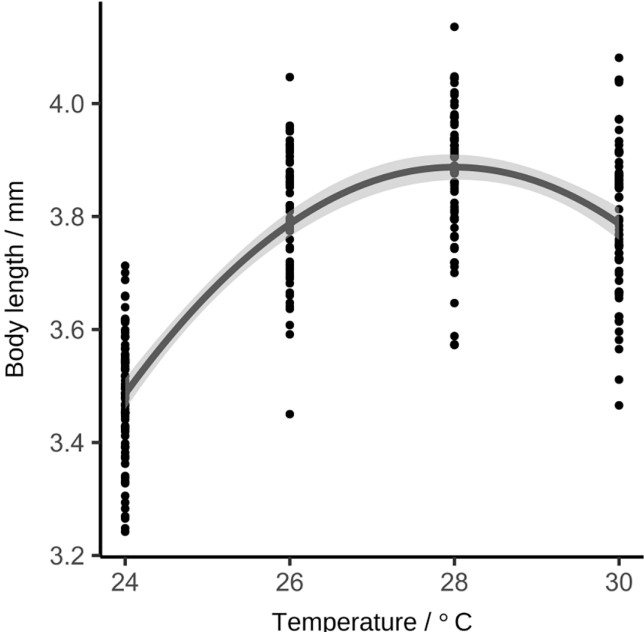

**Fig 5. Temperature dependence of body length.** Relationship between body length and incubation temperature of 119 hpf zebrafish eleutheroembryos modelled using a quadratic model ($R^2 = 0.63$). The shaded grey area represents the 95% CIs for the mean predicted response (Wald-type intervals) Each dot represents the measurement from one larva that was incubated at the respective temperature fixated laterally and measured from the tip of the snout to the end of the tail (excluding the fin) (n = 259 in three repeats).

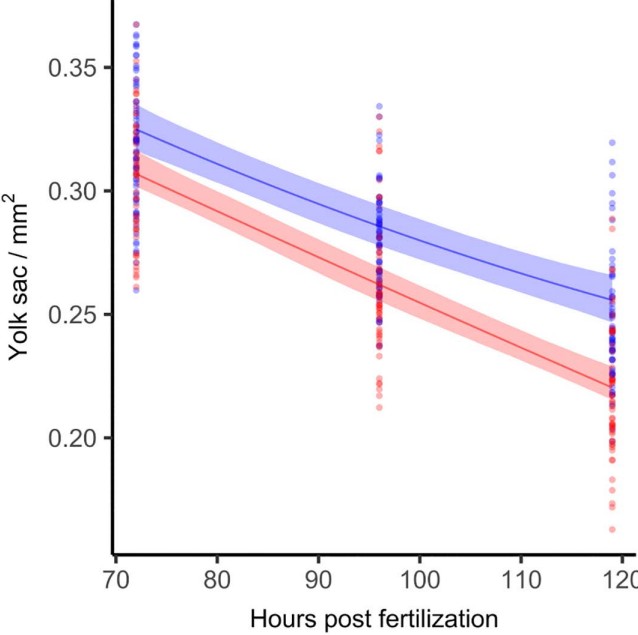

**Fig 6. Yolk sac depletion over time.** Yolk sac size of zebrafish eleutheroembryos over development time in hpf at 26°C (blue) and 28°C (red). A Shape Constrained Additive Model (monotonously decreasing and convex bend) was fitted with a random effect term for replicate affiliation. The shaded areas represent the 95% CIs. Different individuals were fixated laterally to be measured and yolk sac volume approximated by its area ($R^2_{26°C} = 0.59$; $R^2_{28°C} = 0.64$. $n_{26°C} = 176$, $n_{28°C} = 174$ in three repeats; $p < 0.001$, permutation test).

still utilizing yolk reserves. This is corroborated by Wilson [35], who argues that at 120 hpf at 28.5°C, the intestinal tract is still not fully developed, thus limiting the eleutheroembryos' ability to absorb nutrients from exogenous sources efficiently. And finally, Jardine and Litvak [36] report that the yolk sac consumption of zebrafish larvae that were incubated at 28.5°C is only fully consumed only after 165 ± 12 hpf. Taken together, these findings indicate that the yolk sac is the primary energy source of the developing eleutheroembryo until 120 hpf at an exposure of 28°C, and that its role begins to diminish thereafter, as the eleutheroembryo gradually transitions to external feeding. We therefore conclude that eleutheroembryos that are incubated at 26°C should not be classified as "independent feeders" at 120 hpf.

**3.2.2 Eye size.** The initiation of eye development occurs at the 6-somite stage at both incubation temperatures, which is consistent with previous findings [37]. However, the rate of eye growth in eleutheroembryos incubated at 28°C is faster compared to those incubated at 26°C (Fig 7). See S3 Fig for regression coefficients and residual plots. The developmental delay in eleutheroembryos incubated at 26°C is 13 hours at 119 hpf (95% CI 7–19.8, $p < 0.05$).

As observed in many teleost species, eye growth and visual system development continue throughout a zebrafish's life [38]. Therefore, the influence of temperature on eye development likely persists beyond larval stages, potentially affecting long-term visual function and growth dynamics. This would be in line with Scott and Johnston's [39] observation that thermal acclimation capacity of adult zebrafish can be modified by temperature also after early development.

**3.2.3 Summary.** A summary of the developmental delays of zebrafish eleutheroembryos incubated at 26°C, compared to 28°C, is shown in Table 1. We observed a progressing delay between successive endpoints. Over all endpoints at the final development stage (119 hpf) a minimum delay of 7 hours and a maximum delay of 26.3 hours was found.

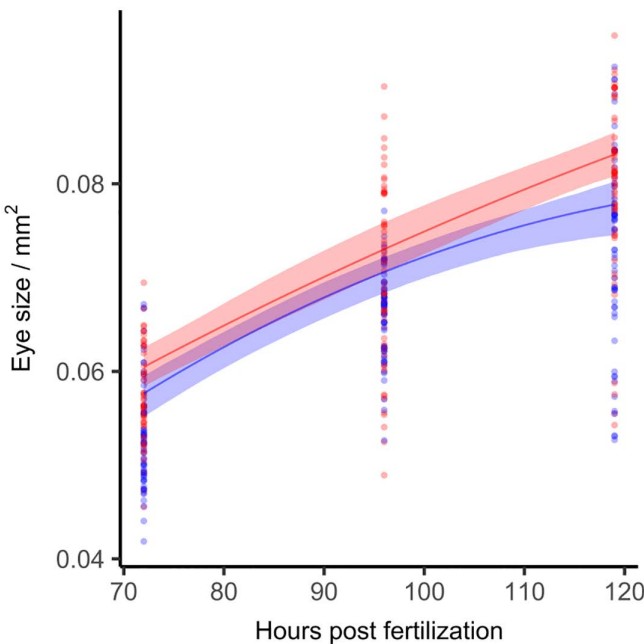

**Fig 7. Eye size development over time.** Eye size of zebrafish eleutheroembryos over development time in hpf at 26°C (blue) and 28°C (red). A Shape Constrained Additive Model (monotonously increasing and concave bend) was fitted with a random effect term for replicate affiliation The shaded areas represent the 95% CIs. Different individuals were fixated laterally to be measured and eye size was determined by area ($R^2_{26°C} = 0.64$, $R^2_{28°C} = 0.62$. $n_{26°C} = 176$, $n_{28°C} = 175$ in three repeats; $p < 0.05$, permutation test).

**Table 1. Summary of the developmental delays of the respective examined endpoints of the zebrafish eleutheroembryos incubated at 26°C when compared to 28°C.**

| | | Hours of delay | | |
|---|---|---|---|---|
| Endpoint | dpf | min | average | max |
| Embryonic Stages | 1 | 2 | 3 | 4 |
| Onset of Heartbeat | 2 | 3.5 | 4 | 5 |
| Hatching | 3 | 3 | 8 | 10 |
| Body Length (high-resolution) | 5 | 15.9 | 18.6 | 21.3 |
| Body Length (low-resolution) | 5 | 7.9 | 12.9 | 17.6 |
| Eye Size | 5 | 7 | 13.3 | 19.8 |
| Yolk Sac Consumption | 5 | 13.4 | 19.8 | 26.3 |

Minimum and maximum delay are given as bootstrapped 95% confidence intervals for measured endpoints at respective day post fertilization (dpf).

## 4. Conclusions

Our study reveals that the sensitivity of zebrafish eleutheroembryos to incubation temperature is more complex than previously recognized. The analyzed endpoints respond differently to the temperature difference. The regulatorily accepted age limit of 120 hpf appears to be based on standard incubation temperatures of 26°C – 28.5°C, but our findings suggest that this limit should be closely linked to a better defined and more narrow temperature range.

### 4.1 Temperature sensitivity and developmental delays

The criteria for independent feeding assembled by Strähle et al. [14] are based on studies using a maximum incubation temperature of 28.5°C. Our study demonstrates a minimum delay of 13.4 hours at the end of the incubation time in yolk sac consumption when comparing zebrafish embryos incubated at 28°C versus 26°C. This estimate is conservative, as the developmental trajectories increasingly diverge over time. This delay is further corroborated by other endpoints, such as body length and eye size, indicating significant sensitivity to temperature variations from early developmental stages.

The conclusions of the present study are based on morphological indicators such as yolk sac depletion, rather than direct assessments of feeding behavior or gut functionality. While the use of such proxies is consistent with earlier work [8,14], the direct evaluation of digestive functions, such as gut motility, enzyme activity or ingestion assays, would provide additional evidence. Future studies incorporating these endpoints would help to further assess the appropriateness of exposure time limits established for animal welfare reasons.

Developmental delays become apparent very early in the life cycle of zebrafish. Markers of early development, such as heartbeat initiation and hatching, show a smaller delay. In contrast, later endpoints, such as yolk sac consumption, show increased delays, which supports the hypothesis that temperature-related differences in development will continue to increase over time. This aligns with the findings of Scott and Johnston [39], who demonstrated that the incubation temperature chosen during early development influences gene expression in adult zebrafish, particularly in genes involved in energy metabolism, muscle contraction and remodeling, and cellular stress.

The high-resolution time series provides highly detailed information, but the resulting data show considerable variation among repeats, likely due to the intense and frequent handling required. In contrast, the low-resolution time series includes three repeats with a large number of individuals, but the wider confidence intervals reflect the lower temporal resolution. More frequent measurements in the low-resolution time series would likely yield narrower confidence intervals, allowing the developmental delay to be estimated more precisely and potentially converge toward the true average observed in this study (~16 hours after five days).

## 4.2 Implications for experimental protocols

Incubating zebrafish embryos at 26°C presents several advantages. The slower development at 26°C allows researchers more time to observe specific developmental stages or endpoints, which is especially valuable for studies focusing on organogenesis or detailed gene expression patterns. On the other hand, the slower pace of development at 26°C might hamper the analysis of late endpoints, and therefore the study of more complex ones like behaviors. This holds especially true, if decision-makers in animal welfare adhere to the 120-hour limit proposed by Strähle et al. [14] without taking temperature into consideration. For instance behavioral endpoints, which are increasingly used in ecotoxicological research become more stable the older the eleutheroembryos are [40]. Based on our findings the current German time limit for zebrafish embryo tests could be extended from 120 hpf to at least 125 hpf for embryos incubated at 26°C (as opposed to the 139.5 hpf estimate made by Strähle et al. [14] with the formula derived by Kimmel et al. [8]). This adjustment accounts for the observed developmental delays and incorporates a conservative safety margin. This seemingly minute extension from 120 to 125 hpf is critically important for the implementation of the LDTT.

Our findings also have implications for another ecotoxicological test, such as the FET test. Here the guideline allows a range from 25°C to 27°C. In light of our results and previous publications of the influence of temperature on toxicity [41,42] we consider this margin to be too large.

Incubating at 28°C accelerates development, making this temperature well suited for high-throughput and large-scale studies. Higher temperatures compromise embryo survival [31] and should therefore be avoided.

The results of the presented study advocate for better acknowledging the links between incubation temperature and the development of zebrafish embryos. We found a minimum 7-hour developmental delay for later endpoints at 26°C compared to 28°C, indicating significant developmental plasticity. This also complicates data interpretation and comparability across studies.

Strain-specific response patterns and laboratory conditions, such as water quality, incubation practices (incubator type, incubation in glass beakers versus well plates), and the handling procedures used for the experimentation (especially the time the embryos spend out of the incubator), significantly influence developmental outcomes. These factors may confound reproducibility and are not yet systematically addressed in most protocols. It is therefore crucial to report on those seemingly inconsequential technical details, in order to ensure transparency and reproducibility of the data published. We also advocate for multi-strain and multi-laboratory studies under harmonized conditions to systematically quantify and assess these differences to ensure comparability and reproducibility between zebrafish studies.

## Supporting information

**S1 Table. Descriptions of the embryonic stages determined during the high-resolution time series.**
(PDF)

**S2 Table. Delays in development observed for different endpoints computed with confidence interval separately for each replicate.**
(PDF)

**S1 Fig. Parameters and diagnostic plots for non-parametric Shape Constrained Additive Model (SCAM with integrated smoothness) with a random effects term for replicate.**
(PDF)

**S2 Fig. Model diagnostics for the regression of body length over temperature with a quadratic model.**
(PDF)

**S3 Fig. Parameters and diagnostic plots for non-parametric Shape Constrained Additive Model (SCAM with integrated smoothness) with a random effects term for replicate.**
(PDF)

**S3 Table. Representative overview of temperature-dependent developmental differences in zebrafish embryos at 26°C and 28°C at three early stages: 14, 18, and 24 hpf.**
(PDF)

**S4 Table. Delay in hours post fertilization for zebrafish embryos to reach developmental stages at 26°C and 28°C.**
(PDF)

**S4 Fig. Relationship between the hours post fertilization (hpf) at different temperatures and the corresponding normalized hpf at 28.5°C, comparing data generated on zebrafish embryo stages by Kimmel et al. (1995) to our data on staging, where comparable.**
(PDF)

**S5 Fig. Relationship between the hours post fertilization (hpf) at different temperatures and the corresponding normalized hpf at 28.5°C, comparing observed data points with predicted values based on fitted regression lines (adapted from Kimmel et al. (1995)).**
(PDF)

**S5 Table. Raw data of developmental rates derived from Kimmel et al. (1995) by using Web Plot Digitizer (Raharzi, 2024).**
(PDF)

## Author contributions

**Conceptualization:** Angelina Miller, Katja Lisa Schröder, Karsten Eike Braun.

**Data curation:** Katja Lisa Schröder.

**Formal analysis:** Angelina Miller, Katja Lisa Schröder.

**Funding acquisition:** Richard Ottermanns, Henner Hollert.

**Investigation:** Angelina Miller, Katja Lisa Schröder, Karsten Eike Braun, Caitlin Steindorf.

**Methodology:** Angelina Miller, Katja Lisa Schröder, Karsten Eike Braun.

**Project administration:** Richard Ottermanns, Martina Roß-Nickoll.

**Supervision:** Richard Ottermanns, Martina Roß-Nickoll, Thomas Backhaus.

**Validation:** Angelina Miller, Katja Lisa Schröder.

**Visualization:** Angelina Miller, Katja Lisa Schröder.

**Writing – original draft:** Angelina Miller, Katja Lisa Schröder.

**Writing – review & editing:** Angelina Miller, Katja Lisa Schröder, Karsten Eike Braun, Caitlin Steindorf, Richard Ottermanns, Martina Roß-Nickoll, Henner Hollert, Thomas Backhaus.

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
