## [Decision Letter · Decision Letter 0]

10 Nov 2025

Dear Dr. Miller,

We look forward to receiving your revised manuscript.

Kind regards,

Hector Escriva, PhD

Academic Editor

PLOS ONE

2. To comply with PLOS One submissions requirements, in your Methods section, please provide additional information regarding the experiments involving animals and ensure you have included details on (1) methods of sacrifice, (2) methods of anesthesia and/or analgesia, and (3) efforts to alleviate suffering.

“This work was funded by the Deutsche Forschungsgemeinschaft (DFG, German Research Foundation) under Germany´s Excellence Strategy – Cluster of Excellence 2186 “The Fuel Science Center” – ID: 390919832.

Funded by the Verein des Hygiene-Institut des Ruhrgebiets e.V., Gelsenkirchen, Germany, funding contract "Standardised and automated evaluation of the zebrafish behaviour test for the environmental toxicological assessment of pollutants" (phase 1 and 2).”

4. Thank you for stating the following in the Funding Section of your manuscript:

“This work was funded by the Deutsche Forschungsgemeinschaft (DFG, German Research Foundation) under Germany´s Excellence Strategy – Cluster of Excellence 2186 “The Fuel Science Center” – ID: 390919832.

Funded by the Verein des Hygiene-Institut des Ruhrgebiets e.V., Gelsenkirchen, Germany, funding contract "Standardised and automated evaluation of the zebrafish behaviour test for the environmental toxicological assessment of pollutants" (phase 1 and 2).”

“This work was funded by the Deutsche Forschungsgemeinschaft (DFG, German Research Foundation) under Germany´s Excellence Strategy – Cluster of Excellence 2186 “The Fuel Science Center” – ID: 390919832.”

5. Thank you for uploading your study's underlying data set. Unfortunately, the repository you have noted in your Data Availability statement does not qualify as an acceptable data repository according to PLOS's standards.

Reviewers' comments:

Reviewer's Responses to Questions

**Comments to the Author**

1. Is the manuscript technically sound, and do the data support the conclusions?

Reviewer #1: Yes

Reviewer #2: Yes

2. Has the statistical analysis been performed appropriately and rigorously?

Reviewer #1: I Don't Know

Reviewer #2: Yes

3. Have the authors made all data underlying the findings in their manuscript fully available?

Reviewer #1: Yes

Reviewer #2: Yes

4. Is the manuscript presented in an intelligible fashion and written in standard English?

Reviewer #1: Yes

Reviewer #2: Yes

Reviewer #1: In this paper, Miller et al characterized the effect of the temperature on the development of zebrafish during the early stages of development and the consequences for laboratory use and animal welfare. They precisely compared the evolution of morphological patterns (body length, eye diameter, yolk sac resorption) and physiological functions (onset of hatching, heart beating) acquired at 26 and 28°C assessed by longitudinal imaging from the 4-cell stage to 120 hours post fertilization and further extend the analysis to 24°C and 30°C from 72 to 120hpf.

The introduction clearly presents the objective of the work in the context of current knowledge in the field and highlights the importance of reassessing the influence of temperature on zebrafish development within the framework of European Directive 2010/63, which defines zebrafish embryos as an alternative to animal experimentation until they are able to feed independently, i.e. around 120 hpf (NC3R). As described by Strähle et al., the assessment of the stage of independent feeding should be based on several criteria: yolk sac consumption, maturation of the digestive organs (opening of the mouth and anus, development of the intestine), free swimming activity and ability to ingest food. As emphasised by the findings of Miller et al., Parichy et al showed that the relationships between size and developmental stage vary with temperature, highlighting the importance of considering developmental progress rather than age when conducting animal experiments on post-hatch embryonic stages.

Miller et al., use a non-linear relationship between fish development rate and temperature, based on the equation developed by Kimmel et al. in order to more accurately extrapolate the incubation period before independent feeding at different temperature conditions.

Major concerns

This study yielded very interesting results for the scientific community, providing new data on the influence of temperature on the development of zebrafish embryos by comparing different temperature regimes (24, 26, 28 and 30 °C) or after exposure to temperature variations (in the case of embryos in the high-resolution time series exposed to frequent removal from the incubator l.325-328).

The exploration of this last point might improve the manuscript by exposing the fish to fluctuating or oscillating temperature regimes in order to describe the influence of these variations more precisely.

Minor concerns

High temporal resolution microscopy described for the study of early stages of development (between 1 hpf and 40 hpf) can induce phototoxicity with adverse effects on living samples, consequences of illumination that can affect developmental dynamics, sample morphology and reproducibility of results. This should be discussed on the same basis as repeated handling of the embryos (l.179), genetic or conditions of incubation (l.247-255).

Malformation and death have been associated at temperature above 32.5°C (l.290-293). Did the authors observe higher mortality rates at 30°C in their experiments? Did any malformations occur? If so, can it be commented on ?

In paragraph 3.1.2, there are discrepancies between the text and the figures (l.294-295. This should be corrected.

In paragraph 3.1.4, the authors showed that the body length of the embryos was smaller when comparing the high and low resolution time series (l.319-320). Adding a short sentence at this level on the batch variation before returning to this point at the end of the paragraph could make the table easier to read. Please also comment the potential phototoxicity effect.

In paragraph 3.2.1, the authors monitored yolk sac consumption 72, 96 and 119 hours post fertilization. Would it be possible to reinforce these results by providing information on the opening of the mouth and anus and/or the development of the intestine?

Reviewer #2: This study investigates the temperature-dependent development of zebrafish (Danio rerio) embryos, comparing incubation at 26°C and 28°C. The authors conducted two types of experiments with detailed morphological measurements (eye size, yolk sac area, body length) at 72, 96, and 119 hpf. The key finding is that development is consistently delayed at 26°C compared to 28°C, with the delay increasing for later developmental endpoints. The study concludes that guidelines should link incubation temperature to developmental progress to ensure both scientific rigor and proper animal welfare standards.

The study is well designed, and the results are highly significant and impactful with clear practical implications. In addition, the authors present some limitations and future directions to explore which valorise the work. However, I found some minor issues that need to be addressed:

- L114, how was anaesthesia achieved?

- L116, was a fixative used or an immobilizer? Which solution was used?

- L121, if 53-61 individuals were used from a total of 72, what happened to the remaining individuals?

- L121, “fixated” or immobilized?

- L296, this difference should be included in the figure to better visualize the statistical differences between temperatures. The same applies to the hatching results.

**Do you want your identity to be public for this peer review?** For information about this choice, including consent withdrawal, please see our Privacy Policy

Reviewer #1: No

Reviewer #2: No

---

## [Author Response · Author response to Decision Letter 1]

20 Nov 2025

Please find the Response-to-Reviewers Letter under submitted documents as docx. file. The responses are listed in a table.

---

## [Decision Letter · Decision Letter 1]

17 Dec 2025

Temperature-dependence of early development of zebrafish and the consequences for laboratory use and animal welfare

PONE-D-25-52201R1

Dear Dr. Miller,

We’re pleased to inform you that your manuscript has been judged scientifically suitable for publication and will be formally accepted for publication once it meets all outstanding technical requirements.

Kind regards,

Hector Escriva, PhD

Academic Editor

PLOS One

Additional Editor Comments (optional):

Reviewers' comments:

Reviewer's Responses to Questions

**Comments to the Author**

Reviewer #2: All comments have been addressed

2. Is the manuscript technically sound, and do the data support the conclusions?

Reviewer #2: Yes

3. Has the statistical analysis been performed appropriately and rigorously?

Reviewer #2: Yes

4. Have the authors made all data underlying the findings in their manuscript fully available?

Reviewer #2: Yes

5. Is the manuscript presented in an intelligible fashion and written in standard English?

Reviewer #2: Yes

Reviewer #2: The authors have addressed all my comments and I have no other issues with this work. The work is presented in a fashion way which can have a significant impact in the area

**Do you want your identity to be public for this peer review?** For information about this choice, including consent withdrawal, please see our Privacy Policy

Reviewer #2: No

---

## [Editor Report · Acceptance letter]

PONE-D-25-52201R1

PLOS One

Dear Dr. Miller,

I'm pleased to inform you that your manuscript has been deemed suitable for publication in PLOS One. Congratulations! Your manuscript is now being handed over to our production team.

Kind regards,

on behalf of

Dr. Hector Escriva

Academic Editor

PLOS One